# OpenReview forum: "Visual Token Compression Enhances Model Robustness of VLMs"
_ICLR.cc/2026/Conference — ICLR 2026 Conference Withdrawn Submission_

### Official Review · Reviewer_T5KY · 2025-10-26

**Soundness:** 1
**Presentation:** 2
**Contribution:** 1
**Rating:** 0
**Confidence:** 4

**Summary:**

The paper proposes to prune visual tokens from VLM models to improve their robustness to jailbreak attacks, hallucination, and improve inference speed. They identify the OOD visual tokens using a distance metric and then use that for pruning tokens.

**Strengths:**

Unfortunately, I struggle to find strengths in this paper, please refer to the weaknesses section.

**Weaknesses:**

In the current state the paper has a lot of flaws:
1. It is completely unclear what is the motivation of the paper and why should one choose this approach of visual token pruning towards any goal. Like what is the problem for the solution provided in the paper?
2. And why only prune visual tokens, what is the text is very long, why not prune that? Also how do you prune tokens at a specific layer? Do you set that embedding to zero ?
3. Why is the distance from text embeddings a correct metric for finding OOD visual tokens? What is my text prompt is just blank string or one word string like "describe:" or very short generic string like "what is in this image?" -- why does distance to the text embeddings tell anything about out of distributionness of the visual token embeddings?
4. Are the setting for layer and percentage removal of tokens same across all the experiments or are they different for Table1, Table2, and Table 3?
5. Can you explain how dropping visual tokens increases your performance in benchmarks in Table 3?
6. What are the numbers in Table 3 for these benchmarks: MME (P), MME (C), OCRBench?
7. How does your approach work if I flipped the text and image position, so now text comes before image in a prompt?
8. How does your approach work if I have two images and a text prompt in the middle?
9. How does your approach work if I have no text prompt, just the image (which is not unusual say if my text prompt is typographed in the image)
10. What happens when there are multiple images with text prompt at the end?

**Questions:**

Please refer to the weaknesses section.

---

### Official Review · Reviewer_MThG · 2025-10-26

**Soundness:** 3
**Presentation:** 3
**Contribution:** 2
**Rating:** 4
**Confidence:** 3

**Summary:**

This paper introduces Out-of-Distribution Visual Token Pruning (OOD-VTP), a method designed to enhance both the robustness and computational efficiency of VLMs. The core approach involves pruning r% of visual tokens at layer k by measuring the distance between each visual token and the textual feature space, with the intuition that tokens far from this space may act as out-of-distribution inputs contributing to model vulnerabilities. The hyperparameters r and k are determined through grid search optimization. The method is evaluated on its ability to defend against jailbreak attacks and reduce hallucinations across seven diverse benchmarks, demonstrating an average improvement of 13.46% in jailbreak defense while maintaining competitive performance on hallucination mitigation and general datasets.

**Strengths:**

The proposed method is conceptually intuitive and does not require architectural modifications to existing VLMs, making it readily applicable to various models. The experimental validation is thorough, spanning seven diverse benchmarks and multiple model architectures (including evaluation on jailbreak defense, hallucination mitigation, and general performance), which demonstrates the method's broad applicability and effectiveness.

**Weaknesses:**

1. **Limited Theoretical Validation**: The proposed distance metric uses the maximum distance between a visual token and the set of textual features. This definition lacks mathematical stability and closure, since it is not continuous and does not have a well-defined structure. While I recognize the choice of this definition is based on empirical results, its theoretical basis is not sufficiently discussed. The authors compare the maximum distance and minimum distance criteria and find that the maximum approach performs better. This result is interesting, as it suggests that even when a visual token appears as an outlier under the maximum criterion, there still exist textual features that are very close to it. This observation deserves further analysis.

2. **Incomplete Efficiency Analysis**. While the authors claim notable inference cost reduction, they do not account for the computational overhead of the grid search required to determine optimal values for k and r. Since this optimization must be performed separately for each backbone model and dataset combination, the amortized cost may significantly diminish the practical efficiency gains, especially for practitioners working with multiple models or domains.

3. **Hyperparameter Sensitivity**: The reliance on grid search for k and r suggests potential sensitivity to these choices. The paper would benefit from ablation studies showing performance stability across different hyperparameter settings and guidance on how to select these values efficiently for new models or datasets.

**Questions:**

The proposed method has the potential to reveal which regions or tokens are prioritized by vision-language models. It is unfortunate that the authors did not include any results or discussion in this area.

---

### Official Review · Reviewer_yQEo · 2025-10-26

**Soundness:** 2
**Presentation:** 2
**Contribution:** 2
**Rating:** 2
**Confidence:** 5

**Summary:**

This paper explores the relationship between visual token compression and security vulnerabilities of VLMs and reveals that appropriately evicting certain visual tokens can enhance the robustness of VLMs. The authors design a simple yet effective strategy that prunes visual tokens with the smallest similarity scores to the textual feature space. Extensive experiments across various benchmarks prove the effectiveness of the designed method.

**Strengths:**

1. The proposed method is simple yet effective in enhancing VLM's robustness, as validated by extensive experiments.
2. This work builds a novel perspective for VLLM security, bridging token compression and VLLM robustness.
3. The writing is clear, and the paper is easy to follow.

**Weaknesses:**

1. **Lack of Methodological Novelty**. The proposed method simply consists of a score that decides which tokens to prune, which is more like an extension of FastV and fails to bring any algorithmic innovation or methodological inspiration.
2. **Insufficient clarification about the relationship between the defined OOD tokens and model robustness**. In the paper, the authors suggest that the reduction of these so-called OOD visual tokens can facilitate the robustness of VLMs. However, I do not find any rigorous theoretical analysis or direct experimental evidence. What is the connection between OOD tokens and model vulnerabilities? This is unacceptable if the authors simply attribute this phenomenon to the reduction of 'out-of-distribution' tokens.
3. **Figure 1 is also weird and unsuitable**. The authors aim to present that the multimodal alignment in VLMs is imperfect. However, the relationship between images and texts is essentially a many-to-many mapping, i.e., an image can indeed be described by different texts, and a text can describe various images. This is what Figure 1 actually conveys, rather than the multimodal misalignment in VLMs.
4. **Missing results on POPE benchmark.** It is also necessary to consider POPE analysis in the hallucination evaluation.

Overall, the contribution of this paper is marginal, lacking in-depth analysis regarding the underlying mechanism and methodological inspiration.

**Questions:**

See weaknesses.

---

### Official Review · Reviewer_4y3c · 2025-10-28

**Soundness:** 2
**Presentation:** 2
**Contribution:** 2
**Rating:** 2
**Confidence:** 3

**Summary:**

This paper proposes a method which prunes a subset of visual tokens which are deemed out-of-distribution. This method, not only improves inference efficiency, but also results in significant improvements on factuality and adversarial robustness.

**Strengths:**

## Strengths
- **Strong Empirical Results** The overall results are quite compelling, showing clear improvements on all of the tasks over the baseline.

**Weaknesses:**

## Weaknesses
- Overall, there are many empirical findings presented in this paper, but generally very little explanation is offered for each of them .
- **[Major]** The paper provides limited analysis or intuition for why the proposed OOD-VTP method works. In particular, the rationale behind the distance metric in Eq. 4 is unclear—why is a cross-modal distance (image–text) chosen instead of an intra-modal one (image–image)? Moreover, it is not evident why pruning the farthest visual tokens should meaningfully improve robustness - it seems like this could have an adverse effect in highly compositional images. Finally, the term “out-of-distribution (OOD)” seems somewhat misleading in this context; “outlier” or “misaligned” tokens might be a more appropriate description.

- **[Major]** The paper does not analyze which visual tokens are pruned in practice. Visualizing the pruned tokens—especially in cases where a jailbreak is prevented—would be very interesting to see for a few different cases.

- **[Major]** The paper empirically observes that pruning visual tokens in middle layers yields the best robustness improvements, but provides no explanation for why this is the case. It remains unclear what distinguishes these “robust-pruning layers” from earlier or later layers. A deeper analysis—e.g., probing how token–text correlations or attention entropy evolve across layers—would be very useful.

- **[Major]** The set of baselines is somewhat limited. The experiments primarily compare variants of OOD-VTP and FastV, omitting other training-free pruning approaches that could serve as more informative points of comparison See [1,2] for some representative methods as well as some of the papers that have been cited in the related work section but not included in the experiments.

- **[Minor]** Figure 1 occupies disproportionate space relative to its contribution. The modality misalignment it depicts is already well known, and the figure adds limited value to the paper’s main argument.





**References**
- [1] DivPrune: Diversity-based Visual Token Pruning for Large Multimodal Models, CVPR 2025
- [2] LLaVA-PruMerge: Adaptive Token Reduction for Efficient Large Multimodal Models, ICCV 2025

**Questions:**

See weaknesses

---

### Note · Authors · 2025-11-14

I have read and agree with the venue's withdrawal policy on behalf of myself and my co-authors.